# Selective IgM Deficiency: Evidence, Controversies, and Gaps

**DOI:** 10.3390/diagnostics13172861

**Published:** 2023-09-04

**Authors:** Ivan Taietti, Martina Votto, Maria De Filippo, Matteo Naso, Lorenza Montagna, Daniela Montagna, Amelia Licari, Gian Luigi Marseglia, Riccardo Castagnoli

**Affiliations:** 1Pediatric Unit, Department of Clinical, Surgical, Diagnostic, and Pediatric Sciences, University of Pavia, 27100 Pavia, Italy; ivantaietti@gmail.com (I.T.); martina.votto@unipv.it (M.V.); maria_defilippo@hotmail.it (M.D.F.); matteonaso1992@gmail.com (M.N.); lorenza.montagna@unipv.it (L.M.); daniela.montagna@unipv.it (D.M.); gl.marseglia@smatteo.pv.it (G.L.M.); 2Pediatric Clinic, Fondazione IRCCS Policlinico San Matteo, 27100 Pavia, Italy

**Keywords:** inborn errors of immunity, primary immunodeficiency, hypogammaglobulinemia, infectious diseases, allergy, autoimmunity, selective IgM deficiency, primary antibody deficiency, pediatric immunodeficiency, pediatric IgM deficiency

## Abstract

Selective Immunoglobulin M deficiency (SIgMD) has been recently included in the inborn errors of immunity (IEI) classification by the International Union of Immunological Societies Expert Committee. The understanding of SIgMD is still extremely limited, especially so in cases of SIgMD in the pediatric population. The epidemiology of SIgMD in the pediatric population is still unknown. The pathogenesis of SIgMD remains elusive, and thus far no genetic nor molecular basis has been clearly established as a definitive cause of this primary immunodeficiency. Recurrent respiratory infections represent the main clinical manifestations in children, followed by allergic and autoimmune diseases. No conclusive data on the correct therapeutic management of SIgMD are available. Although, for most SIgMD patients, Ig replacement therapy is not required, it may be recommended for patients with significantly associated antibody deficiency and recurrent or severe infections. Prophylactic antibiotics and the prompt treatment of febrile illness are crucial. There is insufficient evidence on the prognosis of this condition. Therefore, further studies are required to define the disease trajectories and to increase our understanding of the molecular mechanisms underlying SIgMD in order to facilitate a better clinical, immunological, and prognostic characterization of the condition and develop tailored therapeutic management strategies.

## 1. Introduction

Selective Immunoglobulin M deficiency (SIgMD) is a predominantly antibody-affecting deficiency that has been included in the International Union of Immunological Societies (IUIS) classification of inborn errors of immunity (IEI) since 2017 [1,2,3]. According to the European Society for Immunodeficiencies (ESID) criteria, SIgMD is defined by absent or reduced serum immunoglobulin M (IgM) levels in the absence of immunoglobulin deficiencies of the other classes [4]. The significant comorbidities for SIgMD are recurrent infections (even including those with life-threatening severe infections) and an increased frequency of allergic and autoimmune diseases [5]. However, only a few studies have assessed the clinical and immunological features of SIgMD. The pathogenesis of SIgMD remains unclear, and no definitive genetic alterations have been established. SIgMD remains a diagnostic and therapeutic dilemma, especially in the pediatric population, because no conclusive data are available on the correct therapeutic management and the prognosis of SIgMD.

This review aims to summarize the evidence about this apparently rare primary immunodeficiency (PID), focusing on clinical and immunological features, particularly in pediatric patients, to allow for a better understanding of this condition.

## 2. Materials and Methods

A literature search was performed via the online database PubMed by combining the terms “Primary selective IgM deficiency”, “Primary selective immunoglobulin M immunodeficiency”, “Selective IgM deficiency AND children”, “Selective IgM deficiency AND pediatric population”, “Pediatric selective IgM immunodeficiency”, and “gamma-M deficiency AND children”.

The literature review on SIgMD was performed in May 2023, including all publication years. All studies that met the following criteria were included: (i) articles published in English in peer-reviewed journals and (ii) studies wherein the participants were children and adult patients diagnosed with SIgMD. Potentially eligible publications were manually screened and reviewed, and non-relevant publications were excluded.

## 3. Results

The database search found 850 articles. Based on their titles and abstracts, 166 articles met the inclusion criteria. After removing duplicates, 80 articles were analyzed for this review (Figure 1).

### 3.1. Definition (Clinical and Laboratory)

According to the European Society for Immunodeficiencies (ESID) registry criteria, SIgMD is defined as repeatedly absent or reduced serum immunoglobulin M (IgM) levels (less than 2 SD or <10% of the values obtained from healthy controls of the same age or an absolute value <20 mg/dL in pediatric age) with normal levels of serum immunoglobulin A (IgA) and immunoglobulin G (IgG) and IgG subclasses, normal vaccine responses, and the absence of T cell defects (numbers and function) after the exclusion of secondary hypogammaglobulinemia (infections, genetic syndromes, chromosomal abnormalities, drugs, lymphomas, protein-losing enteropathy, nephrotic syndrome, thymoma) [6,7,8] and any other specific IEI. Despite this definition, some authors propose that SIgMD should be defined without the exclusion of the IgG subclass deficiency, alterations in T cell subset numbers and functions, and impaired responses to vaccines in order to better understand the different clinical and immunological phenotypes that lie behind the diagnostic term “IgM deficiency” [9]. In this context, in their 2022 Update of the Classification, the International Union of Immunological Societies (IUIS) Expert Committee defines the only main criteria as the absence/reduction in serum IgM without limiting the definition with additional features (Table 1) [2].

**Table 1 diagnostics-13-02861-t001:** SIgMD definition criteria modified from “Human Inborn Errors of Immunity: 2022 Update on the Classification from the International Union of Immunological Societies Expert Committee” Tangye et al. (2022) [2].

Predominant Antibody Deficiencies				
Isotype, Light Chain, or Functional Deficiencies with Generally Normal Numbers of B Cells				
Disease	Genetic Defect	Inheritance	Immunoglobulin	Associated Features
**Selective IgM** **deficiency**	Unknown	Not established	Low/absent IgM	Pneumococcal/ Bacterial infection

### 3.2. Epidemiology

SIgMD was described for the first time in 1967 by Hobbs et al. in children presenting with meningococcal meningitis [10], and it was considered to be a rare condition [11,12,13,14,15]. It is still mostly ignored as an IEI [16], and no large-scale studies have reported its epidemiology. An unselected community health screening survey reported a prevalence of 0.03% of patients with a complete absence of IgM [17]. More recently, in a screening of more than 3000 healthy adult blood bank donors in Iran, the prevalence of SIgMD was 0.37% [18]. A 0.07–2.1% prevalence in Immunology Clinics has been reported [19]. However, the prevalence of SIgMD in the pediatric population (<18 years of age) is unknown. In children, the reported median age at the onset of symptoms is 3 years, with a median age of diagnosis of 8 years [20] and an average age at the time of diagnosis of 6.0 ± 4.7 years [21]. Male predominance has been reported, along with a variable ratio of male to female patients (7:5 [21], 3:1 [20]).

### 3.3. Etiopathogenesis and Pathophysiology

#### 3.3.1. Evidence from Genetic Studies (Table 2)

No genetic or molecular basis has been established as a definitive cause of SIgMD, and no definitive inheritance pattern has been demonstrated [22]. However, SIgMD has been reported in several chromosomal abnormalities (chromosome 13, 18,19, and 22) [23,24,25,26,27]. The most common association of SIgMD has been with 22q11.2 deletion syndrome [26]. Moreover, SIgMD has been commonly associated with Bloom Syndrome, and it has been reported that preferential damage to IgM production via UV irradiation may be due to the abnormal repair of DNA damage in the lymphoblastoid cell line [28].

Regarding the role of specific genes, no mutations nor deletions have been observed in the *IGHM* gene, including the region encoding the secretory domain of IgM. However, mutations in the *BTK* gene and BCR molecular signaling pathway have been reported in patients with SIgMD [29,30]. Also, BAFFR deficiency has been associated with very low IgM and IgG serum concentrations, as well as very few circulating B cells [31].

**Table 2 diagnostics-13-02861-t002:** Summary of the actual genetic evidence for SIgMD.

Author, Year [Ref]	Type of Study	Genetic Defects	Notes
Seidel et al., 2014 [24]	Case report	Partial trisomy 19p13	Clear reduction in IgM and IgG1 and IgG3 subclasses in a patient with organ malformation.
Inoue CN et al., 2017 [25]	Case report	Trisomy 13	Multi-year history of extensive acne conglobata with abscesses on the face and neck.
Al-Herz et al., 2004 [26]	Case report	De novo chromosome 22q11.2 deletion	A 15-year-old female with velopharyngeal incompetence and developmental and speech delay but no heart defects.
Kung et al., 2007 [27]	Case series	22q11.2 deletion	A 6-year-old boy with recurrent otitis media, sinopulmonary infections, wheezing, velopharyngeal insufficiency, and speech delay. IgM ↓, IgA, and IgG N. Protective antibody titers to protein and carbohydrate antigens.A 14-year-old girl with neonatal seizures, atrial and ventricular septal defects, recurrent otitis media, intellectual disability, and asthma. IgM ↓, IgA, and IgG N. Protective antibody titers to protein and carbohydrate antigens.
Celmeli et al., 2014 [23]	Case report	De novo mosaic ring chromosome 18	
Lim et al., 2013 [29]	Case report	c.347C > T (p.P116L) *BTK* gene mutation	- Six-year-old patient.- X-linked inheritance.
Geier et al., 2018 [30]	Case series	(1) *BTK* E206D mutation(2) biallelic missense mutations in *BLNK*, Pro110Ala, and Ala158Ser	(1) A 15-year-old male with recurrent aphthous stomatitis and recurrent respiratory tract infections (sinusitis, pneumonia, and bronchitis).(2) A 37-year-old male with asymptomatic renal insufficiency (cirrhosis of the left kidney and mild hydronephrosis of the right kidney found at 28 years of age) with no increased susceptibility to infections.
Smulsky et al., 2018 [31]	Case report	Inactivation of the *TNFRSF13C* gene	BAFFR deficiency.

↓ = reduced levels.

#### 3.3.2. Evidence from Immunological Studies (Table 3)

To date, the pathogenesis of SIgMD remains unclear. The selective absence of IgM is hard to explain because most studies reveal normal numbers of circulating surface IgM-positive B cells and normal serum levels of IgG and IgA. Therefore, considering the ontogeny of B cell maturation may indicate normal B cell responses. Conflicting results have frequently been reported by several in vivo and in vitro immunologic, phenotypic, and functional studies conducted over the past 20 years [5]. For these reasons, several studies have focused on T helper cells, regulatory suppressor cells, and the intrinsic defects of B cells [32]. Some in vitro studies have revealed the inability of B lymphocytes to differentiate into IgM-secreting cells due to the insufficient synthesis of secreted Igμ messenger RNA [32,33,34,35].

Moreover, in other studies, IgM-positive B cells produced normal amounts of IgM in vitro when cultured with normal T cells, while T cells showed decreased helper activity for IgM and both for IgG and IgA production. These findings suggest a defect in T helper cell function [36]. A reduction in germinal center (GC) cells and defective specific antibody response were also seen in some patients with SIgMD and mice with defective IgM secretion [37]. Kasahara et al. stated that alterations in follicular helper T (T_FH_) and/or follicular regulatory T (T_FR_) cells might play a role in the pathogenesis of SIgMD, but their role is not clear despite the lower percentage of circulating T_FR_ (CT_F_) cells [38]. Louis et al. demonstrated the potentially pathogenetic role of increased CD8+ Treg in SIgMD [37], and Inoue et al. [39] found increased IgM isotype-specific suppressor T cell activity in patients with SIgMD, as previously reported by Matsushita et al. [40]. Moreover, decreased FcμR expression on marginal zone B-cells may play a role in the pathogenesis of SIgMD [41]. Despite these studies, the specific causative mechanism of IgM deficiency is still unclear. Of note, most data available has been derived from adult patients with SIgMD, and only a few studies include pediatric SIgMD patients [21].

**Table 3 diagnostics-13-02861-t003:** Summary of the immunological pathogenetic evidence for SIgMD.

Author, Year [Ref]	Type of Study	B-Cells Defects	T-cells Defects	Notes
Karsh et al., 1982 [32]	Case report	Intrinsic B cell defect.		
Kondo et al., 1992 [33]	Case reports	Defective secretion of Igμ messenger RNA.		
Ohno et al., 1987 [34] Inoue T et al., 1986 [39] Matsushita et al., 1984 [40]	Case reports		Increased isotype-specific suppressor T cells.	
Yamasaki et al., 1992 [35]	Case-control study	Intrinsic B cell defect.	Decreased T helper cell activity.	
De la Concha et al., 1982 [36]	Case reports		Decreased T helper cell activity.	
Louis et al., 2016 [37]	Case–control study	Increased B-reg.	Increased CD8 T-reg cells.	
Kasahara et al., 2020 [38]	Case–control study		Lower percentage of follicular regulatory T (T_FR_) cells. A higher percentage of circulating follicular helper T (cT_FH_) cells in SIgMD patients with specific antibody response deficiency than in SIgMD patients with normal specific antibody response.	The role is not established.
Gupta et al., 2016 [41]	Case–control study	Decreased FcμR expression on marginal zone B cells.		

T_FR_ cells: follicular regulatory T cells; cT_FH_ cells: circulating follicular helper T cells; FcμR: IgM Fc receptor.

### 3.4. Clinical Manifestations (Table 4)

Recurrent infections represent the presenting manifestation in more than 80% of patients with SIgMD [5,42,43]. Upper respiratory tract infections (including rhinitis, otitis media, and sinusitis) and pneumonia (also recurrent with the possibility of developing bronchiectasis [44]) represent the most common clinical manifestations in SIgMD patients [45,46]. However, invasive infections like septic arthritis [47] and severe life-threatening infections (bacterial meningitis and sepsis) have been reported [43,48,49,50]. Some of the most common microbial organisms include *Streptococcus pneumoniae*, *Hemophilus influenzae*, *Neisseria meningitidis*, *Pseudomonas aeruginosa* [48], *Aspergillus fumigatus*, and *Giardia lamblia* [51]. In children, infectious agents are also represented by *Pneumocystis carinii*, *S. aureus*, *Salmonella* sp, CMV, and *Molluscum contagiosum* [52].

*Mycobacteria* infections have also been described in association with SIgMD [46]. Hassanein et al. identified a case of miliar tuberculosis in a 31-year-old man [53]. Of note, as reported by Consonni et al., SIgMD may be associated with a severe clinical course of *Mycobacterium* infection compared to other healthy children [54]. Other infectious manifestations, such as skin infections [20,21,55,56,57], multiple recurrent hordeola [58], chronic gastritis, cholecystitis, and epididymitis [15], have been reported in these patients.

Several patients (up to almost 40% of patients with SIgMD) display allergic manifestations [20,21], and the frequency of asthma and allergic rhinitis in SIgMD ranges from 30 to 45% [5]. Likewise, autoimmunity and autoimmune diseases are more frequent in patients with SIgMD than in the general population [59].

Goldstein et al. reported autoimmune diseases in 14% of patients with SIgMD [19], although, in children, they are less frequent [46]. Autoimmune diseases, including systemic lupus erythematosus [60], Hashimoto thyroiditis [61], autoimmune thrombocytopenia [62], autoimmune glomerulonephritis [62], autoimmune hepatitis [63], juvenile idiopathic arthritis [64], and rheumatoid arthritis have been described as comorbidities in SIgMD. Also, chronic recurrent multifocal osteomyelitis [65], rheumatic heart disease, psoriasis, and scleroderma [15] have been reported. Lim et al. reported lupus-like nephritis and proteinuria in a 6-year-old patient with a mutation in the *BTK* gene [29]. Like other IEIs, hemophagocytic lymphohistiocytosis (HLH) is reported to be a potential complication of SIgMD [66].

Some cases of neoplastic disease have been reported, particularly in the adult population, while in children, neoplastic diseases are anecdotal [20,67,68,69,70].

Various investigators have also reported a failure to thrive in a few pediatric patients [21,36].

**Table 4 diagnostics-13-02861-t004:** Clinical manifestations in SIgMD.

**Infectious Manifestations**	
Upper respiratory tract infections Recurrent otitis media Sinusitis (recurrent, chronic) Bronchitis Pneumonia (also recurrent) Bronchiectasis [15,44] Urinary tract infections Diarrhea, gastroenteric infections, hepatitis, and cholangitis Lymphadenopathy	Severe infections (meningitis, osteomyelitis, septic arthritis, and deep tissue and liver abscesses) and sepsis (mainly meningococcal and pneumococcal infections; *Pseudomonas*). Mycobacteria infections (also miliar tuberculosis [53] and atypical mycobacterial adenitis [54]). Soft tissue infections and skin infections (also herpes infections, acne conglobate [25], disseminated molluscum contagiosum in a 16-year-old girl [55], recurrent Staphylococcal pyoderma in two adult men [56], and recurrent impetigo in a 6.5-year-old boy [57]. Multiple recurrent hordeola (reported in a 10-year-old boy [58]).
**Allergic Manifestations**	
Allergic rhinitis Asthma and recurrent wheezing in the infancy	Idiopathic angioedema and anaphylaxis (reported in adulthood) Atopic dermatitis
**Autoimmune Manifestations**	
Addison’s disease Autoimmune glomerulonephritis Autoimmune hemolytic anemia Autoimmune thrombocytopenia Celiac disease Crohn’s disease Hashimoto’s thyroiditis Rheumatic heart disease (reported [15])	Myasthenia gravis Polymyositis Idiopathic Juvenile Arthritis and Rheumatoid Arthritis Sjogren’s syndrome Systemic lupus erythematosus Vitiligo Psoriasis and scleroderma (reported [15])
**Neoplastic Manifestations**	
Acute myeloid leukemia, tubular adenoma in the sigmoid colon, and neuroblastoma [20]. Multiple myeloma, non-Hodgkin lymphoma, thyroid cancer, and oropharyngeal carcinoma [68]. Gastric cancer [68]; EBV+ gastric adenocarcinoma in a 53-year-old male with collagenous gastritis and a history of asthma, allergic rhinitis, recurrent upper respiratory tract infections, multiple cases of pneumonia, acute sinusitis, and meningitis [69].	MGUS [68]; IgAλ MGUS in a 21-year-old female with a history of recurrent urinary tract infections [70]. Primary cutaneous anaplastic large-cell lymphoma in a 13-year-old boy [67].

MGUS: Monoclonal gammopathy of undetermined significance.

### 3.5. Immunological Characterization (Table 5)

As previously mentioned, several authors have reported additional immunological abnormalities in patients initially identified with SIgMD.

IgG subclass deficiency has been reported in a subset of SIgMD patients with a rate between 25% [19] and 42% [71], particularly IgG4 subclass deficiency [72,73].

T cell number and function are normal in most SIgMD patients [34,35,37,74]. However, alterations in subsets of patients have been reported [36]. In their study, Lucuab-Fergurur et al. reported that 10% of their adult subjects had reduced CD3+ T cells, 30% had reduced CD4+ T helper cells, and 2% had reduced CD8+ T cells [68].

B cells are normal in most SIgMD patients. However, low B cells have been reported in a few patients with SIgMD [43,68,74]. Non-switched memory B cell values were significantly lower in a subgroup of SIgMD patients than in healthy controls [75,76]. Mensen et al. found reduced class-switched memory B cells, but naïve B cells were normal in their cohort [77].

The lymphocytic proliferation response to mitogens and antigens is maintained in most patients with SIgMD [78]. Raziuddin et al. reported CD4+ T cell deficiency and defective interleukin 2 receptor expression and production by the patient’s peripheral blood lymphocytes in response to mitogenic stimulation. Thus, impaired T cell function may be responsible for IgM-deficient antibody production [79]. Yamasaki et al. previously described a reduction in the proliferative response of patient B cells to *Staphylococcus aureus* Cowan strain I (SAC). Considering in vitro immunoglobulin production, IgM production could be normal [74] or reduced. At the same time, IgG and IgA have been reported to be normal (using a T cell-independent culture system) [35], as reported by Karsh et al. [32]. Considering the response to polysaccharides and proteic antigens, Lucuab et al. reported an unprotected/impaired anti-*Streptococcus pneumoniae* antibody in 47% of patients and an inadequate response to tetanus toxoid in up to 12% of patients in their cohort [36,68].

**Table 5 diagnostics-13-02861-t005:** Immunological findings.

**Immunoglobulins**			**IgG Subclasses**						**Lymphocyte Subsets**			
**IgM**	**IgA**	**IgG**	**IgG1**	**IgG2**		**IgG3**		**IgG4**	**CD3**	**CD4**	**CD8**	**CD19**
↓ or absent	N	N	N or ↓	N or ↓		N or ↓		N or ↓	N or ↓	N or ↓	N or ↓	N; ↓ or absent
**B-cell Subsets**				**In Vitro Ig** **Production**				**Lymphocyte** **Stimulation**		**Response to Vaccines**		
**Naïve** **(CD27-, IgM+, IgD+)**	**Non-switched Memory** **(CD27+, IgM+, IgD+)**		**Class Switched Memory** **(CD27+, IgD-, IgG+ or IgA+ or IgE+)**	**IgA**	**IgM**		**IgG**	**Mitogen**	**Antigen**	**Polysaccharide**		**Protein**
N	N or ↓		N or ↓	N	N or ↓		N	N	N	N or impaired		N or impaired

N: normal. ↓: low/reduction. Ig: immunoglobulin.

### 3.6. Therapeutic Interventions (Table 6)

No conclusive data on the correct therapeutic management of SIgMD are available. Although, for most SIgMD patients, Ig replacement therapy (IgGRT) is not required [71], it may be recommended for patients with significantly associated antibody deficiency or recurrent or severe infections [5,80]. It is also interesting to observe that Goldstein et al. reported that IgGRT in patients with SIgMID and bronchiectasis might lead to a reduced risk of pulmonary infection with consequently better infection control and a preventive role in further progressive bronchiectasis [81]. Notably, it is conceivable that IgM-enriched immunoglobulin replacement therapy with biologically active IgM can potentially prevent bacterial respiratory infections. However, further studies must be conducted to determine the potential role of IgM-enriched IgRT [82]. Prophylactic antibiotics and the prompt treatment of febrile illness are crucial. Adult patients receive more frequent antibiotics and/or IgGRT [20,68] compared to the pediatric population [21]. Vaccines, including pneumococcal and meningococcal vaccines, should be given as scheduled for healthy populations. However, antibody response may be decreased, and conjugate vaccines may require repeated doses to provide protection (if any). However, considering the immunological abnormalities associated with SIgMD, the impairment of T cell function should be excluded before the administration of attenuated vaccines.

**Table 6 diagnostics-13-02861-t006:** Summary of therapeutic interventions that may be beneficial for SIgMD deficiency.

Intervention	Notes
Vaccination	Before the administration of attenuated vaccines, an evaluation of T cell function is advised.
Prompt treatment of febrile illness	
Immunoglobulin replacement therapy	Patients with significant antibody deficiency, particularly in the presence of impaired pneumococcal antibody responses, recurrent or severe infections and/or bronchiectasis.
Prophylactic antibiotics	Particularly in patients with other associated immunological defects.
Management of atopic diseases	May be helpful in reducing the incidence of complicating sinopulmonary infections.

Hib: *Haemophilus influenzae* b.

### 3.7. Prognosis

Although, in some infants, SIgMD may be transient [43], insufficient evidence is available on the prognosis of this condition. In particular, no conclusive data have been collected regarding the course of the disease and the possible evolution of SIgMD to common variable immunodeficiency (CVID) [83].

## 4. Discussion

SIgMD has been recently included in the IEI classification by the IUIS Expert Committee [1,2,3]. However, as recently reported by the US National Institutes of Health (https://rarediseases.info.nih.gov/diseases/12547/selective-igm-deficiency (accessed on 29 May 2023)), our understanding of this condition requires improvement.

The definition of SIgMD is still controversial. Although specific ESID criteria are available, some authors propose that SIgMD should be defined without the exclusion of the IgG subclass deficiency, alterations in T cell subset numbers and functions, and impaired response to the vaccine in order to better understand the different clinical and immunological phenotypes that lie behind the diagnostic term “IgM deficiency.” Indeed, as demonstrated by Janssen et al., only a limited number of SIgMD-diagnosed patients reported in the literature adhere to the ESID criteria [7]. From a clinical perspective, according to the Jeffrey Modell Foundation’s “Four stages of testing” [84], patients usually undergo quantitative IgG-, IgM-, and IgA-level dosage as the first step in the immunological workup. From the available evidence, it is clear that patients exhibiting absent/reduced serum IgM levels after this first stage of testing should undergo a complete immunological workup to identify other possibly associated immunological abnormalities.

According to our literature review, the main clinical manifestations are represented by infections, especially respiratory infections, followed by allergic manifestations [85,86] and autoimmune diseases [20,21,68,87]. Neoplastic diseases have been described with a non-conclusive association with SIgMD.

Considering the probably pathogenetic defects associated with SIgMD reported in the literature, some of them are of particular interest, namely BTK gene mutation [29,30], BLNK gene mutation [30], BAFFR deficiency due to TNFRSF13C gene mutation [31], and Igμ deficiency [33]. Some of these are also causes of agammaglobulinemia. In this category of IEI, we have the BTK deficiency X-linked agammaglobulinemia (XLA), the autosomal recessive μ heavy chain deficiency, and BLNK deficiency [2]. These conditions are characterized by severe bacterial infections, a severe reduction in all serum immunoglobulin isotypes, and profoundly decreased or absent B cells (CD19+ less than 2%) [88]. These features are partially shared with SIgMD, reinforcing the etiopathogenetic hypothesis that it could be determined by a B-cell intrinsic defect. BAFF receptor deficiency due to TNFRSF13C gene mutation is one the IEI showing a CVID phenotype (defined as an age-specific reduction in the serum concentrations of IgG in combination with low levels of IgA and/or IgM and a poor or absent response to immunizations and/or absent isohemagglutinins and/or low switched B cells (<70 percent of age-related normal value) and the absence of profound T cell immunodeficiency and absence of any other defined immunodeficient state [89,90]) characterized by low IgG and IgM serum levels with a variable clinical expression [3]. This consideration strengthens the possible evolution of SIgMD to CVID [83] and the need for a close follow-up for all patients with SIgMD.

Regarding the immunological characterization, according to ESID criteria [4] and Janssen et al.’s definition [7,46], “truly” selective SIgMD has been defined as repeatedly decreased serum IgM levels, normal levels of serum IgG, IgA, IgG subclasses, and normal vaccination responses and the exclusion of T cell defects through the absence of clinical signs suggesting a T cell defect. However, as previously mentioned, several additional immunological abnormalities have been reported in the literature. These findings strengthen the above recommendation for a complete immunological workup in patients with absent/reduced serum IgM levels.

No conclusive data regarding the course of the disease, its potential therapeutic management, and the possible evolution of SIgMD to CVID are available. The most important forms of therapeutic intervention are represented by measures to prevent infections. Both inactivated vaccines (e.g., vaccines against Tetanus, Diphtheria, Pertussis), and live attenuated vaccines (e.g., vaccines against measles, rubeola, mumps, chickenpox, and Rotavirus) are safe in minor antibody deficiencies; thus, they are also safe in SIgMD. On the other hand, in antibody deficiency disorder, the response to active immunization can be highly variable in patients with antibody defects, from cases with an adequate response, as in healthy subjects, to cases in which it is reduced or even absent [91], as reported above. As reported by Ko et al., it is plausible that it could depend on B-memory cell values for polysaccharide antigens in CVID [92]. Because there are several reports of patients with SIgMD with associated T cell defects before administering live attenuated vaccines, it is recommended to check the lymphocyte subsets to inoculate these vaccines safely. In particular, the following immunological parameters are requested: CD4+ T cells ≥500 cells/μL, CD8+ T cells ≥200 cells/μL, and normal T cell response to mitogen (The Center for Disease Control and Prevention recommends even higher CD4+ levels in children below the 6 years of age, namely, CD4+ ≥1000 cells/μL for those between 1 and 6 years of age and ≥1500 cells/μL for those under one year of age)[93].

An accurate extended follow-up (clinical and immunological) in Immunology Clinics, even in asymptomatic patients, is crucial in order to evaluate the evolution of SIgMD in children and adults (persistent or transitory) and identify the possible progression to CVID or other well-defined immunodeficiencies, with a specific focus on patients who showed immune abnormalities in addition to low IgM [20,94].

## 5. Conclusions

Our overall understanding of SIgMD is still extremely limited, especially with respect to the pediatric population. This review evaluated the evidence regarding SIgMD to clarify the condition’s etiopathogenetic and pathophysiologic mechanisms, its main clinical manifestations and immunological features, and its potential treatment options in order to define present and future needs. Further studies, including prospective longitudinal studies with regular immunological evaluation and, if possible, genetic testing, are required to understand the disease trajectories and increase our understanding of the molecular mechanisms underlying SIgMD to facilitate a better clinical, immunological, and prognostic characterization of the condition and therefore develop tailored therapeutic management strategies.

## Figures and Tables

**Figure 1 diagnostics-13-02861-f001:**
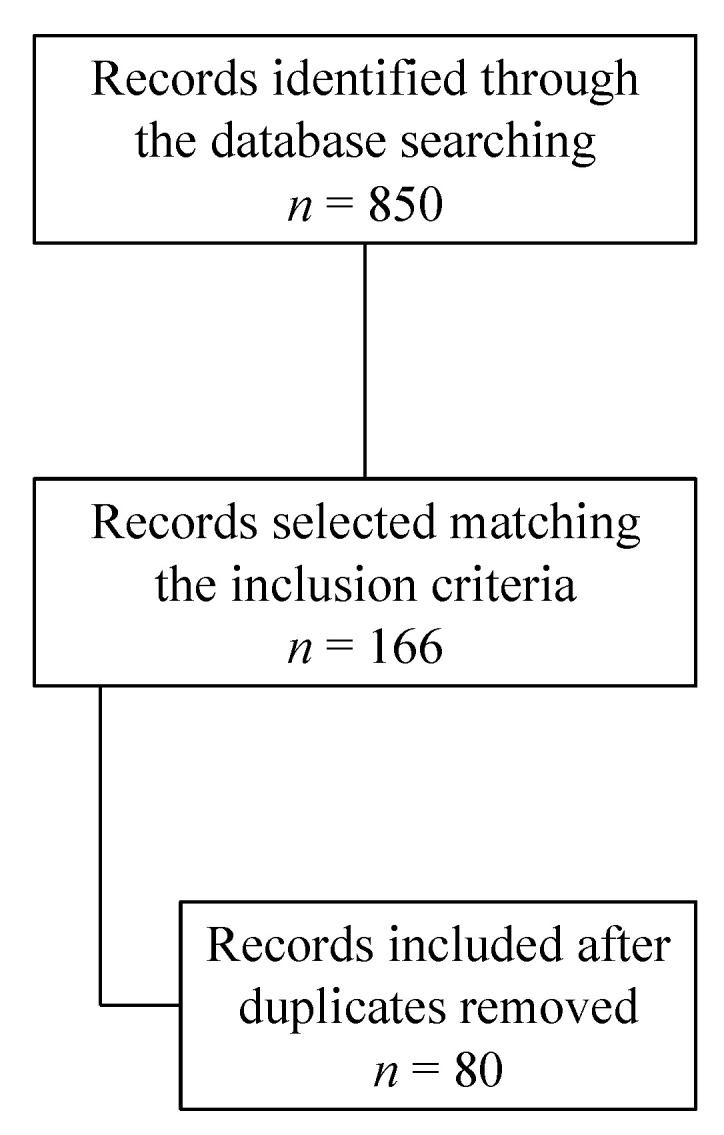
Search strategy.

## Data Availability

Not applicable.

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
