# Peer review of "Selective IgM Deficiency: Evidence, Controversies, and Gaps"

_diagnostics, 2023, doi:10.3390/diagnostics13172861_

Round 1
Reviewer 1 Report
This is an important and timely review article for the selective IgM deficiency (SIgMD) by analyzing 80 previously reported reports. This review well summarizes the genetic, immunological and clinical features of SIgMD, and indicated that SIgMD has not yet been well characterized. English is in general well written. However, I have some comment, which are shown below.
(1) The structures of all the tables and figures are not well prepared. All the tables and figures should be improved. In addition, all the tables and figures should be also placed within the same page, but not be separated into two pages.
(2) "Tables" and "Figures" in the manuscript are inconsistently cited in the text. In addition, some of citations of "Tables" and "Figures" are missing in the text. These should be carefully corrected.
(3) The full spell for ESID is not shown at the first appearance of this abbreviation.
Author Response
This is an important and timely review article for the selective IgM deficiency (SIgMD) by analyzing 80 previously reported reports. This review well summarizes the genetic, immunological and clinical features of SIgMD, and indicated that SIgMD has not yet been well characterized. English is in general well written. However, I have some comment, which are shown below.
(1) The structures of all the tables and figures are not well prepared. All the tables and figures should be improved. In addition, all the tables and figures should be also placed within the same page, but not be separated into two pages.
R: We thank the Reviewer for the comment. We modified the layout of all the Tables in order to facilitate the reading. Moreover, we modified the formatting to fit all the tables into a single page.
(2) "Tables" and "Figures" in the manuscript are inconsistently cited in the text. In addition, some of citations of "Tables" and "Figures" are missing in the text. These should be carefully corrected.
R: We thank the Reviewer for the comment. We revised all the in-text citations to make sure that all the citations for “Tables” and “Figures” are correct.
(3) The full spell for ESID is not shown at the first appearance of this abbreviation.
R: We thank the Reviewer for the comment. We added the full spell for ESID: European Society for Immunodeficiencies. Please see line 33.
Reviewer 2 Report
This review discusses different aspects of Selective Immunoglobulin M deficiency and possible therapeutic strategies. Review is well written but has few issues. Here are my comments:
-Introduction is short, please expand it by providing sufficient background of this disorder.
-Please provide information on other B cell disorders which may be relevant to this disorder such as BTK deficiency, i.e. XLA
-Please discuss how different vaccinations may affect an individual having this disorder.
-Please provide a table for therapeutic interventions which may be beneficial for individuals affected with Selective Immunoglobulin M deficiency.
minor
Author Response
This review discusses different aspects of Selective Immunoglobulin M deficiency and possible therapeutic strategies. Review is well written but has few issues. Here are my comments:
-Introduction is short, please expand it by providing sufficient background of this disorder.
R: We thank the Reviewer for the comment. We added additional details on SIgMD in the Introduction. However, all the clinical and immunological features of SIgMD are detailed in the Results section after clearly showing the literature search methods used.
-Please provide information on other B cell disorders which may be relevant to this disorder such as BTK deficiency, i.e. XLA
R: We thank the Reviewer for the comment. As suggested, we added a paragraph on the B cell disorders. Please see lines 267-284.
-Please discuss how different vaccinations may affect an individual having this disorder.
R: We thank the Reviewer for the comment. We added a paragraph on the vaccinations. Please see lines 295-310.
-Please provide a table for therapeutic interventions which may be beneficial for individuals affected with Selective Immunoglobulin M deficiency.
R: We thank the Reviewer for the comment. We added a table (please see Table 6) to summarize the therapeutic interventions, as suggested
Round 2
Reviewer 1 Report
None.
Reviewer 2 Report
Concerns addressed, no further issues
minor